# Nanoscopic Insight into Water Adsorption and Desorption in Commercial Activated Alumina by Positron Annihilation Lifetime Spectroscopy

**DOI:** 10.3390/ma18163876

**Published:** 2025-08-19

**Authors:** Wojciech Kowalski, Mateusz Kochel, Agnieszka Kierys, Marek Gorgol, Marek Drewniak, Radosław Zaleski

**Affiliations:** 1Faculty of Mathematics, Physics and Computer Science, Maria Curie-Skłodowska University, 1 Maria Curie-Skłodowska Square, 20-031 Lublin, Poland; wojciech.kowalski@mail.umcs.pl; 2Department of Circular Economy, Institute of Energy and Fuel Processing Technology, Zamkowa 1, 41-803 Zabrze, Poland; mkochel@itpe.pl; 3Institute of Chemical Sciences, Maria Curie-Skłodowska University, 3 Maria Curie-Skłodowska Square, 20-031 Lublin, Poland; agnieszka.kierys@mail.umcs.pl (A.K.); marek.drewniak@mail.umcs.pl (M.D.); 4Institute of Physics, Maria Curie-Skłodowska University, 1 Maria Curie-Skłodowska Square, 20-031 Lublin, Poland; marek.gorgol@mail.umcs.pl

**Keywords:** activated alumina, water adsorption and desorption, positron annihilation, drying

## Abstract

Activated alumina is widely used in industry as an adsorbent. Its strong affinity toward water allows for the profound dehydration of gas streams. To optimize such processes, a deeper insight into water interaction with activated alumina is required. This knowledge can be obtained using positron annihilation lifetime spectroscopy, a sensitive tool that unravels previously unknown aspects of adsorption processes. Activated alumina (Compalox^®^ AN/V-813) was subjected to such a study supported by detailed characterization using scanning electron microscopy, X-ray diffraction, and N_2_ adsorption–desorption. A complex porous structure of the material, consisting mainly of boehmite and η-Al_2_O_3_ or γ-Al_2_O_3_, was found. It is responsible for significant differences in adsorption and desorption. The course of adsorption is close to the classical layer-by-layer description. However, there are indications of initial water capture at active sites and final water reorganization consisting of filling the smallest free volumes that remain empty. The narrow mesopore inlets that keep water in the pores even at a relative vapor pressure of 0.4 are primarily responsible for the course of the desorption process. During adsorption, water is mainly maintained in the form of small clusters up to the highest pressures, whereas during desorption, it is continuous until narrow pore openings.

## 1. Introduction

Activated alumina is a porous, inorganic adsorbent distinguished by its high specific surface area, low degree of crystallinity, and abundance of surface-active sites [1]. Structurally, it is partially amorphous and is dominated by mesopores of varied geometry. Due to its amphoteric nature, alumina exhibits both acidic and basic surface properties simultaneously [2]. Its surface is strongly polar, enhancing interactions with a wide range of adsorbates, particularly polar molecules, and contributing to its high adsorption efficiency.

In industrial practice, activated alumina is produced by thermally dehydrating aluminum hydroxide precursors, most commonly gibbsite, boehmite, or pseudo-boehmite. Depending on the calcination conditions and the selected precursor, various crystalline forms may develop, including the stable α-alumina polymorph (corundum) or one of several metastable or transitional phases (η, γ, χ, δ, κ, and θ) [3]. In adsorption applications, the transitional phases γ-Al_2_O_3_, χ-Al_2_O_3_ and, less commonly, η-Al_2_O_3_ are crucial due to their high porosity. Variations in the structure and phase composition of these forms result in distinct adsorption mechanisms, porosity profiles, and surface properties, significantly affecting the material’s interactions with water [4]. In commercial adsorbents, mixed phases and surface groups create chemical and structural heterogeneity. Such complexity translates into excellent versatility in practical use.

Thanks to these properties, activated alumina is widely used in industry as an adsorbent, catalyst, and catalyst support. As an adsorbent, it is used for drying gases and liquids [5,6,7], removing polar contaminants from hydrocarbon streams [8], and desulfurizing liquid fuels [9]. It is also employed in water and wastewater treatment for the adsorption of trace amounts of fluoride [10,11,12], arsenic [13,14], and organic dyes [15,16]. Among these, gas drying is particularly important and widely used. Under typical operating conditions, activated alumina enables pressure dew points of approximately −40 °C [17,18], which, although not as low as those achieved with narrow-pore silica gel (down to −60 °C [19]) or zeolitic molecular sieves of the Linde-type A (LTA) and fujasite-type (FAU) frameworks (as low as −60 to −100 °C [20]), are sufficient for many commercial processes. Importantly, activated alumina offers superior resistance to liquid water, high mechanical strength, and relatively low production costs. These characteristics make it a preferred choice for cyclic adsorption systems such as temperature swing adsorption (TSA) and pressure swing adsorption (PSA). It is particularly well suited for use in air pre-purification units of cryogenic air separation plants, where it typically serves as the initial adsorption layer upstream of 13X zeolite.

Optimizing adsorption processes requires more than a general understanding of the material and involves in-depth knowledge of the adsorbent itself, its surface structure, and the properties and mechanisms governing adsorption and desorption. Activated alumina presents a particularly complex case, with structural and chemical heterogeneity [21,22]. Water adsorption proceeds through a combination of physical mechanisms, including hydrogen bonding, capillary condensation, and specific chemical interactions. Notably, the process initiates at very low relative pressures (p/p_0_). The resulting adsorption isotherm typically shows type IV characteristics with a distinct hysteresis loop [23], reflecting the intricate dynamics of water retention and release.

Hysteresis in adsorption–desorption cycles cannot be explained solely by capillary condensation or irreversible evaporation. Surface-related effects and the clustering of water molecules also play a significant role [21]. Adsorption may proceed via coordinative and dissociative mechanisms [24]. Surface hydroxyl groups contribute to the formation of additional, loosely bound water layers, which may eventually lead to condensation in larger mesopores [25]. Pore filling and emptying is a nuanced process governed by both the surface’s chemical character and interactions between adsorbate and surface, many aspects of which remain not fully understood [26]. Unraveling these mechanisms requires a broad analytical perspective and the use of complementary characterization techniques.

In this context, several well-established methods are widely employed to investigate the structure and properties of porous materials, including activated alumina. These include, among others, low-temperature nitrogen adsorption/desorption isotherm analysis [27,28], FTIR spectroscopy [29], X-ray diffraction [30], nuclear magnetic resonance [31,32], transmission electron microscopy [30], and scanning electron microscopy. Each technique offers a different perspective on the material’s properties but also has its limitations. As a result, a comprehensive understanding of adsorption phenomena often requires combining multiple analytical tools. Classical approaches, such as low-temperature nitrogen adsorption and mercury intrusion porosimetry, remain the standard practice for pore structure analysis. Nitrogen-based measurements of pore filling and emptying can provide valuable structural insights. However, their accuracy at the lower limit of pore sizes is sometimes insufficient. In addition, interactions of nitrogen molecules with the adsorbent may distort the results, for example, by artificially opening pores that would otherwise remain closed [28]. Altogether, such limitations emphasize the need to broaden the scope of characterization methods and to integrate complementary techniques for a more reliable interpretation of experimental data [26].

In recent years, positron annihilation lifetime spectroscopy (PALS) has emerged as a highly sensitive and versatile tool for probing a broad spectrum of porous materials. These include porous systems with diverse free volume characteristics, such as zeolites [33,34,35,36,37], metal-organic frameworks (MOFs) [38], porous polymers [39,40], various forms of silica-based materials [41,42], and composites [43,44]. Positrons exhibit exceptional sensitivity to various free volumes, i.e., spaces with zero electron density. This includes closed and open pores, interfacial regions, and voids where ortho-positronium (o-Ps) can be trapped. A key advantage of PALS is its ability to perform in situ measurements under changing process conditions without altering the sample or disturbing the experimental conditions [45,46,47,48], particularly in systems with water trapped in the pores of clays [49], glass [50], ceramics [51], silica [52,53,54], titania [55], and MOFs [56]. This makes it particularly well suited for precisely characterizing porous materials and extracting reliable data on process flow. When combined thoughtfully with complementary methods, such as X-ray diffraction or scanning electron microscopy, PALS offers insights into adsorption mechanisms that are often inaccessible through conventional approaches.

Although the scientific literature on alumina and its metastable phases is extensive, fundamental studies on adsorption and desorption mechanisms at the nanoscale remain relatively scarce, particularly for commercial adsorbent materials. These materials are typically complex mixtures of multiple crystalline phases, making their analysis significantly more challenging. Furthermore, they often contain various additives derived from proprietary manufacturing processes, complicating the interpretation of experimental findings. Despite widespread industrial use and continued academic interest, many aspects of their surface structure and adsorption behavior remain only partially understood [31,57]. A deeper insight into interactions between water molecules and the surface of activated alumina could help improve the optimization of adsorption processes in practical, real-world industrial applications. Therefore, this work provides insight into the details of water adsorption in a commercially available used activated alumina.

## 2. Materials and Methods

### 2.1. Material

Activated alumina (Compalox^®^ AN/V-813, Martinswerk GmbH, Bergheim, Germany [58]) in the form of low-dust grains with sizes of 1–3 mm was obtained from ABC-Z System EKO s.c., Katowice, Poland. The grains were powdered in a mortar before all experiments.

### 2.2. Methods

The microstructure of the activated alumina powder surface (after gold-palladium sputtering) was observed using a scanning electron microscope (SEM, FEI Company, Quanta 3D FEG, Hillsboro, OR, USA) working at an accelerating voltage of 5 kV. An energy-dispersive X-ray spectrometer (EDS, EDAX TSL AMETEK, Mahwah, NJ, USA)) coupled with the same microscope, working at 20 kV, was used to determine the elemental composition of the sample.

The X-ray diffraction (XRD) pattern of powdered Al_2_O_3_ was recorded using an Empyrean X-ray diffractometer (PANalytical, Malvern, UK) equipped with CuKα radiation (λ = 0.154 nm). The data were collected in the 2θ range from 20° to 100° and referenced against the ICDD PDF4+ 2023 database. The structure was refined by Rietveld analysis using the ‘HighScore Plus—Rietveld Refinement’ ver. 3.0e (3.0.5) software from Malvern PANalytical [59], and the calculated R-factors (R_p_—profile R-factor, R_wp_—weighted profile R-factor, and R_e_—expected R-factor) are given.

The parameters characterizing the porosity of the powdered activated alumina were determined from N_2_ adsorption–desorption at −196 °C using a volumetric adsorption analyzer (ASAP 2020 V4.01, Micromeritics, Norcross, GA, USA). Before the measurement, the sample was degassed at 200 °C for 1000 min. The specific surface area (S_BET_) was calculated based on N_2_ adsorption data using the standard Brunauer−Emmett−Teller (BET) equation [27], while the total pore volume (V_p_) was estimated from single-point adsorption at a relative pressure of about 0.99. The pore size distribution (PSD) was determined from the N_2_ adsorption branch of the isotherm using the Density Functional Theory (DFT) method. The DFT analysis was performed using the Micromeritics MicroActive ver. 1.01 software, assuming nitrogen adsorption on an oxide surface and alternatively a cylindrical or slit pore shape.

### 2.3. Water Adsorption and Desorption Procedure

Adsorption and desorption of water on the activated alumina were monitored with positron annihilation lifetime spectroscopy (PALS). The description of the measurement principle, apparatus, and analysis methods of PAL spectra is presented in Appendix A.

The pressure control was assured by the setup schematically presented in Figure A1 (Appendix A). After placing the sample and positron source in the measurement chamber, the spectrometer was started. The chamber pressure was reduced to below 1 Pa using a rotary vane pump and then further decreased to ca. 10^−4^ Pa with a turbomolecular pump for thorough cleansing at room temperature. This process took about 3 days, as indicated by the stabilized PAL spectra. The chamber was then isolated, and distilled water vapor (Milli-Q, 5 MΩ cm at 298 K) was introduced through a gas gauge. Before introduction, the water was degassed three times using freeze–pump–thaw cycling.

The chamber was incrementally filled with water vapor, with pressure levels increasing every 12 h by ca. 0.05 p_0_ (saturated water vapor pressure at ambient temperature, 295 K ± 1 K). The “inlet” gas gauge was dynamically regulated through software for 30 min at the start of each step, assuring a constant pressure during water adsorption. Then, after closing the valves, a pressure drop was observed for some time, indicating ongoing adsorption. The PAL spectra from the time of valve opening and pressure stabilization were discarded. After reaching p_0_ and stabilizing the PAL spectra, the water reservoir was sealed off. Desorption was initiated by decreasing the pressure with the “outlet” gas gauge (through a fully open “inlet” gas gauge) in 12 h intervals each time by ca. 0.1 p_0_, analogously to adsorption. When the pressure stabilization level became <0.03 p_0_, continuous vacuum pumping was initiated using a rotary pump. After the next 40 h, when the pressure decreased below 1.5 Pa, a turbomolecular pump was started and continued pumping for ca. 29 h.

Finally, after completing the desorption run, the chamber with the sample was filled with water vapor in three steps at different pressure levels and then, analogously, evacuated in three steps, with manual control over the gauges. During this procedure, the PALS spectra were collected to verify if the measured pressure was affected by air leakage during the previous long-lasting adsorption–desorption cycle.

## 3. Results and Discussion

### 3.1. Characterization

The SEM micrographs of the surface of activated alumina powder (Figure 1) show that the milling of Compalox^®^ AN/V-813 grains results in the formation of small, irregularly shaped pieces. Their structure appears to be composed of plate-like forms or sheets that are relatively tightly packed but that easily fall apart when grinding, breaking into smaller fragments. The EDS spectrum of activated alumina powder (Figure A2, Appendix A) contains peaks that can be attributed to oxygen, aluminum, sodium, and carbon. The calculated amount of oxygen is (52 ± 2) wt%/(60 ± 4) at%, aluminum (39 ± 3) wt%/(27 ± 3) at%, sodium (0.29 ± 0.03) wt%/(0.24 ± 0.02) at%, and carbon is approx. (8 ± 4) wt%/(13 ± 6) at%. It is worth noting that the above elemental composition should be treated qualitatively and not quantitatively; in particular, the amount of light elements, including carbon, cannot be precisely determined by SEM-EDS. The presence of sodium is consistent with the producer information [58]. A carbon (C) peak is always visible, even though a specimen under study does not contain carbon, due to the carbon film that is used as a carrier in SEM-EDS. However, considering the XRD result (Figure 2), the presence of carbon should not be treated solely as an artifact. Although the method of production of Compalox^®^ activated alumina is not disclosed and still is secret know-how, it cannot be ruled out that carbon is formed because of contamination from used chemical reagents, including precursors or some organic additives. The addition of such agents was indicated, for example, in patents [60,61,62].

Determining the phase composition of activated alumina was not an easy task, as the investigated material turned out to have a complex composition. The XRD pattern is dominated by diffraction reflexes at 2θ values of 14.5°, 28.2°, 38.3°, 48.9°, and 55.2°, which correspond well with the pattern of aluminum oxyhydroxide γ-AlOOH (boehmite, PDF4+ no. 04-010-5684). The boehmite (which accounts for ca. 64%) is accompanied by alumina, which has a cubic crystal structure with the Fd-3m space group, as relatively small reflexes can be distinguished at 2θ values of 37.7°, 45.9°, as well as at 66.9°, and most importantly, at 19.4°. The presence of the latter signal is a subtle difference in the XRD powder patterns of γ- and η-Al_2_O_3_ but makes it possible to distinguish these two phases from each other, as postulated by Zhou and Snyder [63]. Therefore, this additional phase, which constitutes about 33%, could be considered eta-alumina (η-Al_2_O_3_, PDF4+ no. 04-007-2615). However, considering the complex nature of the XRD pattern of the investigated activated alumina, the presence of a gamma-alumina (γ-Al_2_O_3_) phase instead of η-Al_2_O_3_ cannot be excluded. Additionally, the precise crystalline structure of the γ-Al_2_O_3_ phase remains a topic of scientific debate [64,65], with various structural models proposed, including a cubic structure. Nevertheless, the assumption of only boehmite and η-Al_2_O_3_, or only boehmite and γ-Al_2_O_3_, in the investigated sample does not provide a satisfactory fit to the measured XRD. A much better fit is obtained by additionally assuming the presence of cubic carbon (PDF4+ no. 04-024-9928) and a second additional orthorhombic phase (for example, PDF4+ no. 04-015-9125). The presence of the orthorhombic phase, whose content is below 3%, is evidenced by a broad and low-intensity diffraction signal at a 2θ value of 42.5°. The presence of carbon cannot be ruled out, considering the EDS results and those provided by the manufacturer; however, its presence should be treated with great caution, and its content is certainly low (less than 1%).

Compalox^®^ AN/V-813 can be regarded as a highly porous adsorbent as it has a specific surface area of 320 m^2^/g and a total pore volume of 0.25 cm^3^/g (Table 1). The measured S_BET_ value is consistent with that declared by the manufacturer, while the V_p_ is slightly lower [58]. Moreover, these values are somewhat higher than those reported for many activated alumina prepared by various methods [13], including a precipitation method [16] or the low-heat solid-phase precursor method [30]. The higher S_BET_ and V_p_ values compared to other activated alumina are most likely related to the high boehmite content in the investigated Compalox^®^ AN/V-813 activated alumina.

The nitrogen adsorption/desorption isotherms are shown in Figure 3a. The adsorption isotherm is a type II isotherm of the International Union of Pure and Applied Chemistry (IUPAC) classification [28], with the hysteresis loop most closely resembling the H4 type, suggesting the coexistence of micropores and large mesopores in the activated alumina structure. A sharp step-down in the desorption branch at about p/p_0_ ∼ 0.45 can be attributed to cavitation-induced evaporation. It can, therefore, be assumed that there are free volumes in the activated alumina under study in the shape of an “ink bottle” with a narrow neck and a wide body part, i.e., pores whose interior is larger than their narrow entrances.

The information on the PSD was derived from the adsorption branch of the isotherm by applying the DFT method with an assumption of either cylindrical (Figure 3b) or slit shape (Figure 3c) of pores. Regardless of the pore shape, the obtained PSD indicates the presence of a group of small mesopores with sizes below 5 nm (PSD with the maximum at 2.9 nm or 3.2 nm when the cylindrical shape or slit shape of pores is assumed, respectively). The DFT method that assumes a cylindrical pore shape corresponds to the model used in the PALS method, which allows for a comparison of the results from both methods. However, considering the phase composition of the sample, it seems more appropriate to assume a slit-like shape of the pores in the DFT model.

### 3.2. Water Adsorption and Desorption

The maximum entropy lifetime analysis (MELT, Appendix A) of the dry-activated alumina spectrum (Figure 4a, relative water vapor pressure p/p_0_ = 0) reveals the existence of six components with different lifetimes. They originate from para-positronium (p-Ps, lifetime < 0.2 ns), positrons that annihilate without forming a Ps state (e^+^, lifetime ca. 0.4 ns), and ortho-positronium (o-Ps, four components with lifetimes > 1 ns). The large number of o-Ps components indicates a complex porous structure, with the PSD of two long-lived components (Figure 5a, p/p_0_ = 0) corresponding to the distribution obtained from N_2_ adsorption/desorption isotherms and presented in Figure 3b. The shift of positron porosimetry results toward smaller sizes indicates that the parameter value Δ = 0.166 nm may be too low for characterizing activated alumina pores, and model calibration is desirable, as it was done for different materials in refs. [66,67]. However, since the main subject of this work is the study of water adsorption, for which this parameter value is the most appropriate [68], it will be retained in further calculations of positron porosimetry results. The intermediate-lived components, i.e., with lifetimes of the order of several nanoseconds, correspond to smaller (sub-nanometer) free volumes in the activated alumina structure. They most likely originate from the trapping of o-Ps at the crystal grain boundaries, whose diversity and significant concentration are expected due to the coexistence of different phases in the material (Figure 2).

The trend of changes in individual components of PALS spectra was followed by comparing the results for a series of spectra at different stages of water adsorption (Figure 4a):dry (p/p_0_ = 0),approximately in the middle of mesopore filling (p/p_0_ = 0.50),pores almost filled (p/p_0_ = 1.00),and then desorption (Figure 4b):empty mesopores start to appear (p/p_0_ = 0.44),approximately in the middle of mesopore emptying (p/p_0_ = 0.30),pores almost completely emptied (p/p_0_ = 0.02).

The p-Ps’ lifetime changes significantly with the relative pressure of water p/p_0_ (i.e., with the amount of water adsorbed). In the dry material, it has a value of 0.17 ns (Figure 4a), which is quite a large value for porous oxides. A value greater than 0.125 ns, expected for intrinsic p-Ps annihilation, may indicate an admixture of positron (not forming positronium) annihilation in the dislocations of the alumina crystal structure [69]. During adsorption, this value gradually increases to over 0.21 ns, which is characteristic of water [70], where numerous radiation processes occur that affect the lifetime of p-Ps [71,72]. During desorption, an analogous change occurs in the opposite direction.

Also, the lifetime of long-lived (compared to alumina defects) positrons not bound in Ps changes with the relative pressure of water. However, in this case, the increase in its lifetime from 0.40 ns to over 0.45 ns is visible only when long-lived components (related to mesopores) disappear. This is also observed in both adsorption and desorption cases. The change in this lifetime may also be related to radiation processes, which do not affect the lifetimes of positrons as long as the thermalizing positron releases energy in water clusters separated from those in which its annihilation occurs [73], i.e., until pores are filled with a continuous water volume. It is also possible that this change results from a distortion of this component due to the inclusion of part of the grain boundary annihilation, as described later.

The changes in the o-Ps’ lifetimes (>1 ns) and the shape of their distributions are easier to discuss after transforming them into PSDs (Figure 5). However, it should be remembered that such an operation results in minor discrepancies from the actual distributions, which arise from the simplified assumptions that all pores have the same cylindrical shape, and their walls are made of the same material (while they may be covered with water). An easily noticeable and reversible change is that the peaks of the size distribution in the sub-nanometer size range (D < 1 nm) come closer together during adsorption and move apart during desorption. The decrease in the size of the micropores, from almost 1 nm to 0.5–0.6 nm, can be simply explained by the location of more and more water molecules (which are smaller than 0.3 nm) within them, which reduces the space available for o-Ps. The explanation for the increase in the size of the space at the grain boundaries from 0.3 nm to 0.4 nm is somewhat more complex. The origin of this component changes as the volume of water in the pores increases. Individual water clusters become so large that o-Ps bubbles can form inside them. The bubbles are spaces that normally do not exist in liquids but are created as a result of the equilibrium between the surface tension of the liquid and the zero energy of o-Ps, which “pushes” the surrounding molecules [74]. Annihilation at grain boundaries still occurs, but its probability is much lower than that of annihilation in bubbles. It can no longer be isolated as a separate component due to its small intensity and a similar lifetime to the “bubble” component. The lifetime and size of the bubbles in fully hydrated activated alumina (Figure 4a and Figure 5a, p/p_0_ = 1) are slightly smaller than in bulk water. This indicates the proximity of the walls, which limits the growth of bubbles and/or the interaction of the walls with water molecules. This latter is particularly likely due to the numerous -OH groups in boehmite. Both sub-nanometer free volume groups return to their initial sizes upon desorption. It is worth noting that during desorption (change between p/p_0_ of 1.0 in Figure 5a and 0.44 in Figure 5b), a visible decrease in the number of o-Ps bubbles (area under the peak at 0.40 nm drops ca. two times) at a constant micropore volume (area under the peak at 0.55 nm remains appx. constant) is visible. This means that the micropores are the last to empty.

The already wide and complex mesopore size distribution is further widened during water adsorption (Figure 5a, p/p_0_ = 0.50). The apparent bimodal PSD is shifted toward smaller sizes (1.8 nm → 1.4 nm and 3.8 nm → 2.2 nm). This suggests that the pore walls are covered with a layer of water, which limits the free space inside them. This differs from the course in similarly sized cylindrical silica pores [53], where “plugs” were formed instead of a layer of water on the walls. This may be due to the predominance of the forces binding water to the pore walls over the forces of interaction between water molecules. The appearance of an additional very wide peak with a maximum of about 9 nm, covering nearly the entire mesopore range, is responsible for the broadening of the whole pore distribution. Its appearance can be attributed to the reduced probability of o-Ps trapping in smaller pores and the higher efficiency of positronium formation on the surface covered with a water layer than on alumina. In turn, after filling the pores with water, a small peak remains in the PSD around 2.4 nm. It probably comes from a small group of closed pores. Given the low intensity of the component from which it originates, both its position and width are subject to very high uncertainty.

The evolution of the mesopore size distribution during desorption proceeds in a classical manner. First, next to the closed pores with a size of ca. 1.9 nm, the largest pores (D > 10 nm with the PSD maximum at 15 nm) appear (Figure 5b, p/p_0_ = 0.44), the distribution of which widens (D > 5 nm with the PSD maximum at 11 nm) with a decrease in the relative water vapor pressure (p/p_0_ = 0.30). At the same time, free volumes appear in pores whose walls are covered with water, as shown by the restored bimodal distribution with peaks at 1.5 nm and 2.5 nm. As a result, the distribution is very similar to that during adsorption (Figure 5a, p/p_0_ = 0.50) but at a much lower pressure, p/p_0_ = 0.30. At the end of water removal (p/p_0_ = 0.02), the PSD resembles the initial one in dry activated alumina, but it is much broader. It can be suspected that this result is due to the inaccuracy of the MELT analysis, which does not reproduce broad distributions well [75]. However, it is also possible that the narrow distribution observed before water introduction was due to pore size averaging by migrating positronium in the interconnected pore network [76]. In this case, the wide pore distribution would mean that despite the low p/p_0_, not all connections between mesopores were unblocked.

To follow the course of adsorption and desorption, PALS spectra measured at successive pressures, changing with a small pressure step, were investigated. The analysis of lifetime distributions would not be legible due to the large number of results. Therefore, a simplified analysis model was used, assuming the presence of 5–6 components with lifetimes without distributions. This significantly reduced the uncertainty of the results and made it easier to track changes in the spectra with the water vapor pressure. Two short-lived components, which correspond to (1) p-Ps and (2) positrons that have not formed Ps, do not provide additional information beyond that discussed in the analysis of the MELT results (i.e., distributions of lifetimes and free volumes). Therefore, we will focus only on components 3–6 originating from the o-Ps annihilation. Since the formation, trapping, and annihilation of o-Ps, which are considered in the positron porosimetry models, can be influenced by various additional effects (e.g., the already mentioned change in the material forming the pore walls [77] and o-Ps migration [78]), converting lifetimes and intensities to pore sizes and volumes could be misleading. Therefore, further interpretation of the adsorption and desorption process will be carried out using primary parameters obtained from the PALS spectra, i.e., lifetimes (τ_n_) and intensities (I_n_), where n = 3, 4, 5, 6.

The results of the analysis using a simplified PALS spectra model (Figure 6) show that the most distinct changes are visible in the intensities of the individual components, which is why they will be mainly used to discuss the observed process. It should be noted that these changes cannot be observed based only on water adsorption isotherms, which have a smooth shape without any characteristic inflection points [58]. The sixth component provides information about the largest pores. It is usually assumed that its intensity (I_6_) is primarily related to the total pore surface area or concentration at a given pore size. However, it may also depend to some extent on connections with smaller pores, from where additional o-Ps may flow, as well as the difference in the probability of o-Ps formation on free volume walls composed of different materials (e.g., η-Al_2_O_3_, γ-AlOOH, water). During adsorption, the change of I_6_ occurs in almost the entire pressure range, ranging from the predominant value in the spectrum of about 13% to almost zero. However, the main sigmoidal decrease is observed in the p/p_0_ range between 0.3 and 0.7. An analogous change in the opposite direction occurs during desorption, but it is sharper and covers a lower and much narrower range of p/p_0_, i.e., 0.4–0.2. The simplified model provides imprecise information about τ_6_, as the dispersion of its values exceeds any noticeable trend.

Otherwise, for the fifth component, the decrease in lifetime is clear. This confirms the observations presented during the analysis of MELT results, i.e., covering the walls of several-nanometer-sized mesopores with water. Whereas the different course of I_5_ from I_6_ is somewhat surprising. Already in the pressure range of 0–0.1, it is almost halved from its initial value of about 10%. This decrease in I_5_ can be attributed to the occupancy of active sites, which significantly participate in the formation of positronium, e.g., functional groups, with water molecules. Such an effect may be absent in the case of component 6 because pores from group 5, with dimensions of 2 nm, occur in a different form of material (see Figure 2) than larger pores from group 6. However, it should be remembered that o-Ps annihilating in the large mesopores of group 6 are most likely formed in micropores, which most likely have a different wall structure (e.g., due to their small size). Interestingly, the low-pressure effect of increasing I_5_ does not occur during desorption, which may indicate a permanent change in the character of the pore surface due to interaction with water.

At p/p_0_ > 0.1, I_5_ stabilizes, while I_6_ systematically decreases. This may indicate that the small mesopores of group 5 have blocked inlets and are inaccessible to water below a certain threshold pressure, which can push condensed water through the blocked inlets. This is consistent with the “ink bottle” pore shape postulated when discussing the low-temperature nitrogen adsorption. However, it is also possible that these pores are too large for capillary condensation in this p/p_0_ range, while “micropore driven” I_6_ decreases due to condensation (or rather “clogging”) in micropores even smaller than those of group 5. The term “capillary condensation” may be misleading for possibly slit pores of this groupc. This shape would additionally justify relatively high condensation pressure for group 5 mesopores, i.e., p/p_0_ of about 0.5, above which I_5_ decreases almost identically to I_6_. Above p/p_0_ of 0.8, the intensities of both components are so small that they cannot be distinguished with satisfactory accuracy, and a single component is used to approximate them both. During desorption, although the increase in I_5_ occurs in parallel with I_6_, it maintains a significantly lower value in contrast to adsorption. This discrepancy between I_5_ and I_6_ is consistent with typical capillary evaporation, when large mesopores of group 6 are emptied before small mesopores of group 5. However, the increase of I_6_ occurs at much lower pressure than its decrease during adsorption. This can be explained if some of the large pores of group 6 can be emptied only through the narrow inlets of small mesopores of group 5. Nevertheless, some shifts in the emptying pressure can also be related to the different shapes of group 5 and 6 pores.

In the micropores, represented by the fourth component, the size of the voids also changes significantly with increasing pressure, as reflected by a significant decrease in lifetime (τ_4_). Interestingly, this change very closely reproduces the changes in I_6_ in both the adsorption and desorption cases. This is a premise to consider that the migration of o-Ps [41] formed in the larger micropores is the main source of the high I_6_, i.e., o-Ps annihilating in the largest free volumes. This provides an alternative explanation for the previous interpretation of the shift of the increase in I_6_ toward low pressures during desorption due to the narrow inlets of group 5 pores. The decrease in τ_4_ during adsorption is accompanied by a gradual increase in I_4_, from about 2% to its doubling at p/p_0_ of about 0.5. Then, I_4_ increases up to 15%, followed by its stabilization in the p/p_0_ range of 0.7–0.8, and then decreases to zero at the saturated vapor pressure. This change can be attributed to the relative nature of the o-Ps intensities, i.e., the initial increase in I_4_ could result from the decrease in I_5_ and I_6_, while the decrease is due to the predominant probability of o-Ps formation in water (i.e., I_3_, which will be discussed later). However, comparing this result with earlier studies on water and alkane adsorption in silicas and polymers [53,77,79], it can be expected that this is an effect of the formation of a gap at the interface between water and alumina. The surprising difference from the other materials is that the size of this gap is much larger in alumina. Again, this suggests a significantly different interaction of water with alumina (understood as boehmite/η-Al_2_O_3_/γ-Al_2_O_3_). This can also indicate a significantly different (rougher) surface topography, which may consist of micropore inlets and grain boundaries. The gap is greatly reduced at p/p_0_ > 0.9 after the rearrangement of water molecules or a change in the chemical nature of the alumina surface. The latter effect may be consistent with the lack of low-pressure changes in I_5_, which were observed at the beginning of adsorption and at the final stage of water removal. During desorption, the I_4_ increase effect, which can be observed at p/p_0_ of about 0.35, is much weaker, if present at all.

The interpretation of the shortest-lived third o-Ps component is quite unambiguous compared to other components. As already mentioned, its initial origin from the free volumes on the grain boundaries changes to o-Ps bubbles. Such bubbles can only form when the water clusters have a sufficiently large volume. Therefore, this component does not convey information about the presence of water molecules in the sample, but rather about the formation of its clusters larger than the o-Ps bubble, i.e., 0.5 nm. Nevertheless, the third component carries very clear and useful information about the course of water adsorption and desorption, as well as the differences between the course of water condensation and evaporation. During adsorption, three distinct ranges can be identified, each with an approximately linear increase in I_3_, albeit at a different rate. Below a relative pressure of about 0.5 p/p_0_, there are practically no water clusters, and the increase in I_3_ results rather from a decrease in the other intensities. In the range of 0.5–0.9 p/p_0_, the volume of water capable of containing an o-Ps bubble increases approximately linearly. This indicates the filling of subsequent pores, the merging of small water clusters into larger ones, etc. Finally, the rapid increase in I_3_ at p/p_0_ > 0.9 most likely indicates the merging of separated water volumes, but also the disappearance of space allowing o-Ps to locate at the water–solid interface.

The course of desorption is fundamentally different from that of adsorption, reflecting hysteresis not so much in the amount of water but in the transition from its dispersed to continuous form and vice versa. For most of the pressure range, down to p/p_0_ of about 0.4, I_3_ decreases very slightly, indicating that confined water remains in a continuous form with almost no loss. Below this pressure, a relatively rapid drop in I_3_ occurs. At p/p_0_ of about 0.25, it reaches a value characteristic of free spaces at grain boundaries, indicating that water has dispersed and is unable to accommodate o-Ps bubbles. This type of very wide hysteresis seems to confirm the ink bottle shape of pores. Comparing the pressure values at which the I_3_ drop occurred with the PALS desorption study in silica [53], the pore inlet sizes can be estimated at 2–3 nm. This would point out the group 5 pores, but the different interaction of water with alumina than with silica may distort this value. Finally, important information about the pore structure is provided by the lack of bubble size growth (i.e., τ_3_) during desorption, which would be expected due to the occurrence of negative pressure in the pores [80]. Such behavior is characteristic of strongly connected pores with narrow inlets, which confirms the previous interpretation of the course of the adsorption and desorption process.

## 4. Conclusions

Standard methods of material characterization indicate that the structure of the Compalox^®^ activated alumina is relatively closely packed and composed of plate-like forms or sheets. It consists mainly of boehmite, accompanied most probably by η-Al_2_O_3_ or γ-Al_2_O_3_. However, the presence of a small admixture of other phases cannot be excluded. Both micropores and large mesopores are present in the structure of activated alumina, arranged in a manner that facilitates cavitation-induced nitrogen evaporation, such as the ink bottle structure.

Positron annihilation lifetime spectroscopy provided a unique insight into the course of adsorption and desorption. Although the interpretation of the results is not direct and requires reconstructing the phenomena leading to the annihilation of positronium, placing it in the context of previous research allows us to understand details of the course of the studied processes that are otherwise inaccessible.

The general course of water adsorption in activated alumina was determined by positron porosimetry at several characteristic water vapor pressures. Unlike in cylindrical silica pores, where water “plugs” are formed during adsorption, a layer of water forms on the walls of alumina, limiting the free space inside the pores. This may indicate a different ratio of the force binding water to the pore walls and the force of interaction between water molecules in alumina than in silica. Desorption, like adsorption, proceeds in the manner described by the classical picture. The largest empty spaces appear first, and their distribution expands as the relative water vapor pressure decreases. However, positron porosimetry allows us to conclude that at the end of desorption, not all connections between mesopores are unblocked.

In turn, monitoring adsorption and desorption by PALS in fine pressure steps reveals additional details of water confinement in the pores of activated alumina. Initially, water molecules adsorb on active sites in smaller mesopores. This suppresses o-Ps formation, testifying that these sites are rich in weakly bonded electrons. Further capillary condensation in smaller mesopores occurs at a relative water vapor pressure characteristic of the pore size, considering that their shape may resemble slits. The correct interpretation of PALS results is possible thanks to the knowledge that o-Ps, while annihilating in the largest mesopores, most likely forms in much smaller micropores. On this basis, a picture emerges that during adsorption, micropores are systematically blocked already from low pressures, resulting in the inhibition of the migration of o-Ps from micropores to the largest mesopores. However, the micropores are only filled when water is pushed into them just below the saturated water vapor pressure. This may be due to the strong binding of water to the functional groups close to their inlets. Then, the water also merges into a continuous volume, previously remaining mainly in the form of isolated clusters. This can be particularly interesting in technical applications, e.g., requiring a large surface area of water, and it is difficult to determine using any other method than PALS. This continuous volume of water is present over most of the pressure range during desorption, with no signs of cavitation. The subsequent changes in free volumes suggest that larger mesopores can only empty through relatively narrow inlets that become unblocked only at low pressures.

The material collected in this study can be used to optimize the use of Compalox^®^ activated alumina as it provides novel insights into the mechanisms of water adsorption and desorption within the porous structure of activated alumina, unveiling details that are inaccessible to conventional characterization methods. The utilization of PALS facilitates in situ observation of these sorption processes at the laboratory stage itself. Although PALS is a relatively costly technique, it delivers critical information early in the development process, often at a considerably lower cost than that of pilot-scale testing. This approach facilitates informed decision-making in the selection and optimization of composite adsorbents or the exploration of new applications for activated alumina at the early stages of process and material design, thereby enhancing both the efficiency and cost-effectiveness of material development.

However, above all, it is worth emphasizing its importance for further interpretations of water adsorption results on other materials. Similarly, previous water and alkane adsorption studies performed for various adsorbents helped to interpret the results presented in this work. The research presented in this work may also become an inspiration for further exploration of the mechanisms related to the surface chemistry of alumina in the case when the interactions of water molecules with the Al-OH groups (acidic sites) on boehmite and Al^3+^ (Lewis acid sites) on η-Al_2_O_3_ compete with each other.

## Figures and Tables

**Figure 1 materials-18-03876-f001:**
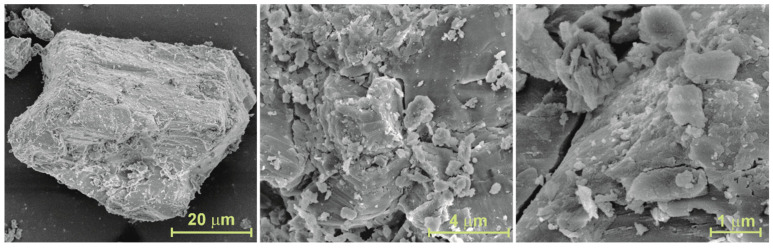
SEM micrographs of activated alumina powder at different magnifications.

**Figure 2 materials-18-03876-f002:**
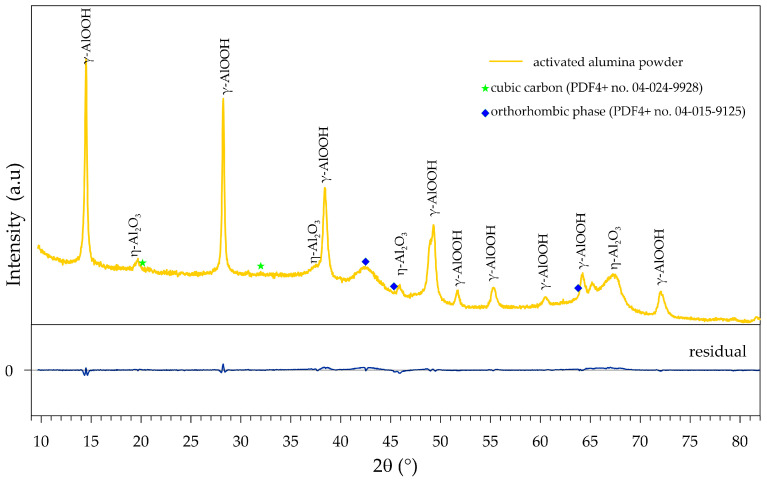
XRD pattern of activated alumina powder. The bottom solid line shows the difference between the calculated and observed intensities (R_p_ = 6.46%, R_wp_ = 9.77%, R_e_ = 2.15%).

**Figure 3 materials-18-03876-f003:**
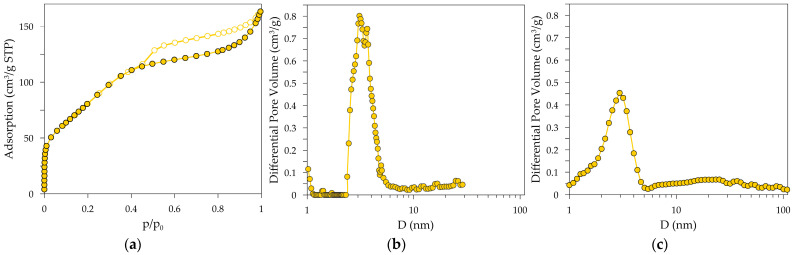
N_2_ adsorption/desorption isotherms (**a**) and pore size distributions of activated alumina powder determined by the DFT method with an assumption of the cylindrical (**b**) and slit shape (**c**) of pores. The lines are provided for convenience.

**Figure 4 materials-18-03876-f004:**
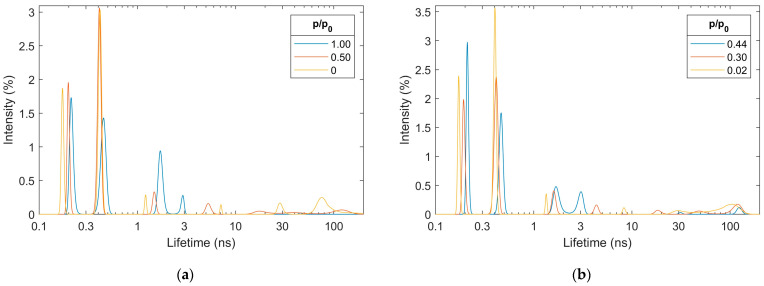
Positron lifetime distribution obtained using MELT for subsequent stages of water adsorption (**a**) and desorption (**b**) from activated alumina powder.

**Figure 5 materials-18-03876-f005:**
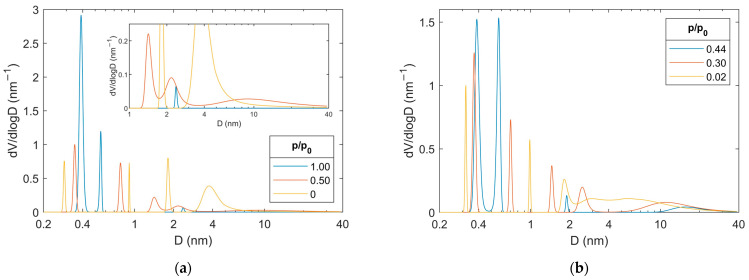
Pore size distributions obtained using positron porosimetry from MELT’s o-Ps lifetime distributions for subsequent stages of water adsorption (**a**) and desorption (**b**) from activated alumina powder.

**Figure 6 materials-18-03876-f006:**
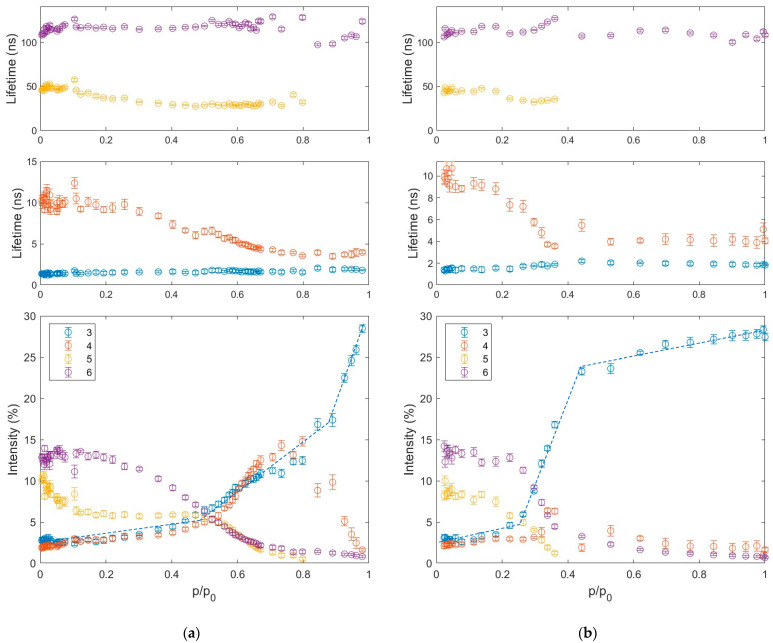
Lifetimes and intensities of o-Ps components (3–6) determined by LT as a function of relative water vapor pressure during adsorption (**a**) and desorption (**b**) from activated alumina powder. Dashed lines are eye guides.

**Table 1 materials-18-03876-t001:** Parameters characterizing the porosity of activated alumina powder obtained from low-temperature N_2_ adsorption/desorption measurements: the specific surface area (S_BET_); the total pore volume (V_p_); the pore diameters at the peak of PSD calculated by the DFT method from the N_2_ adsorption data with an assumption of cylindrical (D_p1_) and slit shape (D_p2_) of pores.

Sample Name	S_BET_ (m^2^/g)	V_p_ (cm^3^/g)	D_p1_ (nm)	D_p2_ (nm)
activated alumina	320	0.25	2.9	3.2

## Data Availability

The original contributions presented in this study are included in the article. Further inquiries can be directed to the corresponding author.

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
