# Peer review of "Nanoscopic Insight into Water Adsorption and Desorption in Commercial Activated Alumina by Positron Annihilation Lifetime Spectroscopy"

_materials, 2025, doi:10.3390/ma18163876_

Round 1
Reviewer 1 Report
Comments and Suggestions for Authors
This study represents the first application of positron annihilation lifetime spectroscopy (PALS) to investigate the water adsorption mechanism of a commercial activated alumina (Compalox® AN/V-813). By integrating PALS with SEM, XRD, and N₂ adsorption characterizations, the dynamic behavior of water molecules within the material’s complex pore structure is elucidated. Nevertheless, the current work exhibits notable shortcomings in the rigor of fundamental material characterization and the depth of data interpretation. A major revision is therefore recommended. The authors are advised to supplement the study with phase quantitative analysis, water adsorption isotherms, and validation of PALS parameters. Furthermore, a more comprehensive mechanistic explanation and a clearer link to industrial applications should be provided. Upon addressing these issues, the research holds the potential to offer valuable insights for the rational design of porous adsorbent materials.
- The XRD results show that the sample contains 64% boehmite and 33% η-Al₂O₃ (Figure 2), but no Rietveld refinement data or goodness-of-fit parameters for quantitative phase analysis (QPA) such as Rwp are provided. Judging the phase ratio merely by visual peak matching lacks rigor, and refined results and residual plots need to be supplemented.
- The pore size distribution (PSD) measured by PALS deviates significantly from the N₂ adsorption results (for instance, PALS shows a 9 nm wide peak while the maximum peak of DFT is only 3.2 nm). The authors attribute this to the uncalibrated Δ parameter (0.166 nm), but no calibration experiments (such as verification with known pore size standards) are provided, which weakens the credibility of the conclusion.
- The author did not provide the water vapor adsorption isotherm (such as data from gravimetric method or quartz microbalance), which led to the lack of a quantitative reference for the adsorption amount in the PALS results. As a result, the inference that "a water layer covers the pore wall at 0.5 p/p₀" cannot be verified.
- PALS observed desorption hysteresis (asymmetry in I₅/I₆ change), but in-situ FTIR or XRD was not used to verify surface hydroxyl reorganization or phase transformation (such as dehydroxylation of boehmite). Additional evidence is needed to rule out the influence of material structural changes.
- The step size of water vapor pressure (0.05 p/p₀/12h) may not have reached equilibrium: the filling of micropores usually requires several days for equilibrium (as in reference 19), and rapid pressure increase may lead to non-equilibrium data. A time-adsorption curve after pressure stabilization should be provided to prove equilibrium.
- The "water layer covering the pore wall" mechanism (in contrast to the "plugging" behavior of silicon-based materials) is not associated with the surface chemistry of alumina: the differential effects of the Al-OH groups (acidic sites) on boehmite and Al³⁺ (Lewis acid sites) on η-Al₂O₃ on water molecules have not been discussed.
- The XRD labels in Figure 2 are confusing (e.g., "cubic carbon" contradicts the main text), and the PDF card numbers and phase names need to be unified.
Reviewer 2 Report
Comments and Suggestions for Authors
This manuscript presents a detailed investigation of water vapor adsorption and desorption mechanisms in commercial activated alumina using positron annihilation lifetime spectroscopy (PALS), supported by classical structural characterization techniques (XRD, SEM, N₂ adsorption). The authors successfully demonstrate how PALS offers nanoscopic insight into pore structure, sorption dynamics, and positronium behavior under varying water vapor pressures.
The study provides valuable mechanistic interpretation of water sorption processes in mixed-phase alumina and highlights the often-overlooked role of pore geometry and wall interactions on desorption hysteresis. The methodology is sound, and the results are clearly presented. However, a few clarifications, comparative contextualizations, and one terminological correction are recommended.
1) Comparative Literature Context: The discussion would benefit from comparison with other advanced materials or methods for handling polar contaminants or dyes, particularly those involving halogenated molecules or ionic compounds.
2) Terminology Correction: In Figures 1 and 2, the Y-axis label uses "absorption (%)", which is misleading in the context of water sorption studies. Please correct it to "adsorption (%)" to accurately reflect the process described.
3) PALS Data Interpretation: The positron lifetime assignments (τ₃–τ₆) and their evolution with vapor pressure are well explained. Nonetheless, a clearer link to macroscopic implications (e.g., drying efficiency, material selection for PSA/TSA units) would enhance applicability.
4) The introduction could be slightly condensed; currently, it repeats core concepts about alumina’s structure in multiple paragraphs.
5) Consider moving part of the detailed PALS parameter discussion to Supporting Information to improve flow.
6) The term "ink-bottle" is used appropriately but could be briefly defined for clarity for non-specialist readers.
This is a scientifically rigorous and well-executed study that successfully applies PALS to explore water–alumina interactions at the nanoscopic level. With a few editorial and contextual improvements, it will make a valuable contribution to the fields of materials science and adsorption physics. Therefore, I recommend its publication after minor revisons.
Reviewer 3 Report
Comments and Suggestions for Authors
The article is well-written and well-structured. The introduction follows a logical sequence. The English is good. The work is innovative. Below are some suggestions:
- Line 217-218 " The calculated amount of oxygen is (52±2) wt% / (60±4) at%, aluminium (39±3)...." - EDS analysis is purely qualitative. I believe it's best to remove these percentages and simply state the elements contained in the sample, or if you want to leave the percentages, make it clear that they are qualitative, not quantitative.
- Line 259: Considering the IUPAC technical report, your isotherm is type II. There is no type 1 and 2 combination. It is clearly type II, and the hysteresis of type h4 will give the porous combinations mentioned. Please correct.
-I believe a table with information on pore volume, diameter, area bet, and other information would be more visual. Also, compare your data with other authors who study activated aluminum.
-Section 3.2. Water adsorption and desorption is very authoritative, where the authors only discuss their own data. It is necessary to discuss the data obtained in comparison with the literature and other authors who have done similar work. Please include more references and discussion comparing the results with yours.
Round 2
Reviewer 1 Report
Comments and Suggestions for Authors
The authors have spent a lot of effort to further improve the manuscript, and they answered all of my questions well. Thus, I would recommend the Editor to consider an acceptance for publication in Materials.